# Understanding Factors Contributing to Vaccine Hesitancy in a Large Metropolitan Area

**DOI:** 10.3390/vaccines11101558

**Published:** 2023-10-02

**Authors:** Paolo Montuori, Immanuela Gentile, Claudio Fiorilla, Michele Sorrentino, Benedetto Schiavone, Valerio Fattore, Fabio Coscetta, Alessandra Riccardi, Antonio Villani, Ugo Trama, Francesca Pennino, Maria Triassi, Antonio Nardone

**Affiliations:** 1Department of Public Health, “Federico II” University, Via Sergio Pansini nº 5, 80131 Naples, Italy; 2General Directorate of Health, Campania Region, Centro Direzionale C3, 80143 Naples, Italy

**Keywords:** vaccination, vaccinations, vaccine hesitancy, KAP model, knowledge, attitude, behaviours

## Abstract

Vaccine hesitancy has become a major global concern, leading to a significant decrease in the vaccination rate, with the World Health Organization recognizing it as one of the top ten threats to public health. Moreover, the health cost generated is evaluated to be 27 billion dollars per year in the US alone. To investigate the association between demographic variables and knowledge, attitudes, and behaviours related to vaccination, a survey-based cross-sectional study was conducted with 1163 individuals. Three models were used to perform a multiple linear regression analysis. In Model I, knowledge about vaccinations was found to be associated with smoking habits, education, and marital status. In Model II, attitudes towards vaccinations were significantly associated with sex, smoking habits, education, marital status, and knowledge. In Model III, behaviours related to vaccination were associated with sex, smoking habits, having children, knowledge, and attitudes. One potential solution to improve behaviours related to vaccinations in the general population is to implement specific public health programs, which can be a cost-effective intervention. This study provides valuable insights into the determinants of knowledge, attitudes, and behaviours related to vaccinations in the general population.

## 1. Introduction

Vaccinations are widely recognized as the most effective measure for preventing infectious diseases [1] with their impact encompassing the reduction in, elimination of, and even eradication of specific diseases [2,3,4,5,6,7]. Furthermore, vaccinations have played a fundamental role in the overall decline in mortality rates over the course of two centuries [8] and are estimated to prevent between 3.5 and 5 million deaths annually [9,10,11]. Nonetheless, most of the recommended childhood vaccinations still fall short of the WHO threshold of 95% [12].

A significant drop in the vaccination rate has been observed in Europe over the past decade, accompanied by an increase in negative attitudes towards vaccinations [13,14,15]. This has led to documented outbreaks of vaccine-preventable diseases (VPDs) [16,17]. Consequently, national health systems have implemented mandatory vaccination policies to tackle this problem [18].

Vaccine hesitancy refers to the refusal of or delay in adherence to vaccination programs and has been recognized as a major threat to global health [19,20]. It is influenced by several factors, including socio-demographic factors, cultural background, lack of knowledge, and low perception of risks [21,22,23,24,25]. The dissemination of unreliable information through the media and online platforms contributes to the erosion of trust in healthcare systems, further fuelling vaccine hesitancy [26,27].

Logistical barriers, including disruptions in vaccine administration due to lockdown orders and difficulties in accessing healthcare during the SARS-CoV-2 pandemic, have contributed to the current low vaccine coverage rates [28]. Despite numerous vaccine information campaigns by national organizations and healthcare providers, vaccine uptake is still heavily influenced by individuals’ own attitudes and knowledge [29]. While behavioural determinants and logistical barriers have been identified, there is a lack of interventions specifically designed to address vaccine hesitancy [30]. Furthermore, few studies have quantified the effectiveness of such interventions in terms of increasing knowledge, improving attitudes, or changing behaviours [31,32].

Recently, numerous studies have been conducted among various population subgroups to analyse knowledge, attitudes, and behaviours related to vaccine hesitancy with a focus on the role of educational attainment and age in adherence to vaccination programs. The literature primarily concentrates on parents, revealing that those with good knowledge and positive attitudes towards vaccinations are more likely to vaccinate their children, especially those with higher levels of education [33,34,35]. Additionally, other studies have been conducted among healthcare providers, indicating that physicians and younger individuals generally possess more knowledge and a better understanding of vaccination recommendations compared to other healthcare workers [36,37,38,39]. However, it is important to note that, due to the impact of the pandemic on academia, most of the recent studies focus specifically on hesitancy related to COVID-19 vaccination, neglecting the importance of other vaccinations [40].

A relative lack of studies conducted in the broader population has been observed. Therefore, the objective of this study is to evaluate the factors that influence vaccine-related knowledge, attitudes, and behaviours. This will be achieved through a comprehensive examination of how variables, such as age, gender, and educational attainment, influence these aspects within a large metropolitan population. The findings from this study will provide valuable insights to guide the development of effective vaccine education and communication strategies. Such efforts can significantly contribute to improving public health outcomes, reducing VPDs, and enhancing overall vaccine acceptance and coverage rates.

## 2. Materials and Methods

### 2.1. Setting and Sample

This cross-sectional study was conducted in Naples, Italy (population 909,048) from December 2022 to March 2023. A snowball sampling method was used to recruit a total of 1670 participants of which 1163 agreed to participate and completed the survey accurately, leading to a response rate of 69.64%; the initial participants were identified and recruited using a snowball sampling method that started with a trusted trial group recommending 2–3 subjects who represented a diverse selection from the general population within the metropolitan area and had expressed interest in participating in the study. Subsequently, we reached out to potential participants through multiple channels, including community organisations, local health clinics, and social media platforms, to ensure a varied and representative sample. Only adults aged 18 years and above and who were residents of the metropolitan area of Naples were deemed eligible for consideration and subsequent inclusion in the study.

The sample size was determined to achieve a margin of error of 3% and a confidence interval of 95%. Based on these parameters, the required sample size was estimated to be 1523 participants, considering a non-response rate of 30%. Thus, the estimated total sample size was adjusted to 1066 participants.

The characteristics of the study population are presented in Table 1. Females accounted for 54% of the participants, and the mean age was 45.46 years, ranging from 18 to 84 years (SD ± 16.72). The majority of the population belonged to the age group over 51 years (40.2%), while those under 30 years represented 24.2% of the sample. Of the respondents, 59.5% reported being in a relationship, and 48.3% stated that they did not have any children. Regarding smoking habits, 56% declared that they do not smoke.

### 2.2. Procedures

Sampling was conducted by researchers on weekdays (Monday to Friday) from 10:00 a.m. to 8:00 p.m. to minimise selection bias. The participants were provided with comprehensive information about the nature, scope, and methodology of the research. Prior consent was obtained from the participants, and they were informed that they had the right to withdraw their consent and discontinue participation at any time without providing a reason. The present study was conducted in accordance with the guidelines and principles outlined in the Declaration of Helsinki, and ethical clearance was obtained according to local legislation.

### 2.3. Data Collection

The questionnaire for this study was developed by a commission of physicians and other healthcare workers, building upon several previously used questionnaires. Any irrelevant or inappropriate questions were removed from the final version. To ensure the questionnaire’s comprehensibility, a beta-test was conducted on a trusted sample of participants whose responses were not included in the study analysis.

The questionnaire consisted of two sections. The first section gathered socio-demographic data and other health-related information, such as gender, age, marital status, parental status, educational level, and smoking habits. The second section comprised 31 questions that assessed the participants’ knowledge, attitudes, and behaviours related to vaccinations. The knowledge and attitude questions were evaluated using three response options, “agree”, “uncertain”, and “disagree”, which were coded as 1, 2, and 3, respectively. The behaviour-related questions offered four response options, “never”, “sometimes”, “often”, and “yes/always”, which were coded as 1, 2, 3, and 4, respectively.

### 2.4. Statistical Analysis

The data collected in this study were analysed using the STATA MP v14.0 statistical software program (College Station, TX, USA). The analysis was conducted in two stages. In the first stage, descriptive statistics were utilised to summarise the basic information of the statistical units. In the second stage, multiple linear regression analysis (MLRA) was performed using three models: Model I, Model II (partially adjusted), and Model III (fully adjusted).

The dependent variables (knowledge, attitudes, and behaviours) were calculated by summing the scores obtained from the corresponding questions (questions with inverse answers were coded inversely). The independent variables were included in all models and consisted of sex (1 = male, 2 = female), age in years, education level (1 = primary school, 2 = middle school, 3 = high school, 4 = university degree), marital status (1 = single, 2 = in a relationship), smoking habits (1 = smoker, 2 = non-smoker), and having children (1 = yes, 2 = no).

In Model II, knowledge was added to the independent variables, and in Model III, knowledge and attitudes were included alongside the independent variables. All statistical tests were two-tailed, and the results were considered statistically significant if the p-values were less than or equal to 0.05.

## 3. Results

Table 2 displays the results of the study’s investigation into the participants’ knowledge of vaccinations. The findings indicate that 50.04% of the respondents correctly identified that antibodies are molecules produced by cells in the human body, while only 38.52% believed that antibodies bind and destroy microorganisms. The study also found that 44.97% of the participants knew that vaccinations stimulate antibody production, and less than half (48.24%) were aware that there are ten compulsory vaccinations in Italy. Moreover, 39.55% of the respondents knew that the tetanus vaccine booster is recommended every ten years in adulthood, while 46.52% knew that the hepatitis B vaccine is mandatory. On the other hand, 24.51% of the respondents incorrectly thought that the HPV vaccine is only recommended for girls. Regarding adverse reactions, only 32.67% of the participants knew that 3 in 100,000 vaccine doses administered result in serious adverse reactions. Finally, 32.67% of the respondents correctly knew that failure to administer vaccinations prohibits enrolment in school, as per D.Lgs. 73/17, the Italian regulation regarding mandatory vaccination issued by the Ministry of Health [41].

Table 3 presents the attitudes of the survey participants towards vaccinations. The findings indicate that a notable proportion, 24.25%, of the respondents view vaccinations as a business, while 39.38% prefer natural remedies over drugs. In addition, 27.34% of the participants agree that vaccinations pose a greater risk than contracting the actual disease, whereas 44.20% disagree. The results also reveal that 30.95% of the participants believe it is necessary for doctors to incorporate alternative medicine into their treatment plans, while 40.33% disagree. Moreover, 28.20% of the respondents think that the only reason to vaccinate their child is to meet the requirements for school or kindergarten.

Table 4 presents the distribution of responses related to vaccine-related behaviours. Regarding recommended vaccinations, approximately 32.9% of the respondents reported having received them, while 33.96% indicated participation in vaccination campaigns organised by local health authorities. Only 39.55% stated that they receive reminders for vaccinations, and 33.10% reported obtaining information about vaccinations when traveling. Merely 34.31% of the participants stated that they receive annual flu shots, and only 29.32% reported vaccinating themselves and/or their children against HPV. Less than half of the respondents, specifically 37.92%, said they educate themselves about the potential risks associated with vaccinations.

Table 5 describes the results of the multiple logistic regression analysis (MLRA) in the three models. In each model, specific references were assigned to the categories: “male” as the reference for sex, “smoker” as the reference for smoking habits, “being single” as the reference for marital status, “have children” as the reference for having children, and “primary school” as the reference for education attainment. In Model I, an association was observed between knowledge regarding vaccinations (used as an independent variable) and non-smoking habits, being single, and higher education levels. Model II showed a statistically significant association between attitudes towards vaccinations and non-smoking habits, being female, being single, and having good knowledge. In Model III, a statistically significant association was observed between behaviours and non-smoking habits, female gender, having children, higher mean scores of knowledge, and positive attitudes.

## 4. Discussion

From December 2022 to March 2023, a cross-sectional survey was conducted in Naples that involved 1163 participants to investigate the limited understanding of vaccine-related knowledge, attitudes, and behaviours. The adult participants completed a 31-question questionnaire, and the collected data were analysed using descriptive statistics and multiple linear regression analysis (MLRA) with three models: Model I, Model II (partially adjusted), and Model III (fully adjusted). The independent variables included in the study were sex, age, education level, marital status, smoking habits, and having children, while the dependent variables were knowledge, attitudes, and behaviours related to vaccinations.

Model I demonstrated a significant association between vaccine knowledge and variables such as non-smoking habits, being single, and higher education levels. Model II revealed a significant association between attitudes towards vaccinations and variables such as non-smoking habits, being female, being single, and having good knowledge. Lastly, Model III identified a significant association between positive vaccine-related behaviours and variables such as non-smoking habits, female gender, having children, higher mean scores of knowledge, and positive attitudes.

Figure 1 suggests a significant association between non-smoking habits and knowledge about vaccinations. While this outcome is unique in the existing literature, it is important to note that there is a lack of research specifically investigating the relationship between non-smoking habits and vaccine knowledge. However, previous studies conducted in South Korea [42,43,44,45] have found that current smokers were less likely to receive vaccinations, which may suggest that smokers have lower levels of knowledge regarding vaccinations. It is, therefore, crucial to conduct further research to better understand the relationship between smoking habits, vaccine knowledge, and vaccine uptake.

Another result of this study was the association between being single and having better knowledge about vaccinations. These findings are in contrast with the only study that assessed such a relation that was conducted among parents attending the Pediatric Emergency Departments in Israel, where being single was associated with lower knowledge [46]. These contrasting findings highlight the need for further research to better assess the relationship between marital status and vaccination knowledge. A possible explanation may be that cultural or societal factors play a role in shaping knowledge towards vaccinations, and these factors can differ across regions and populations.

Furthermore, research has consistently shown a positive association between education attainment and knowledge regarding vaccinations, which is coherent with several previous studies [33,38,47,48] that assessed that college graduates tend to have higher levels of knowledge on vaccinations compared to those with less education. This may be due to several factors, including better health literacy, more exposure to scientific information, and access to resources, such as peer-reviewed journals and academic publications, while individuals from disadvantaged socioeconomic backgrounds may have limited access to education [49,50]. Therefore, promoting education and increasing awareness about vaccinations, particularly among those with lower levels of education, can have a positive impact on both education and health outcomes, improving vaccine uptake and reducing vaccine hesitancy.

The second evidence, as shown in Figure 2, highlights an association between non-smoking habits and attitudes toward vaccinations, which is consistent with precedent studies [42,43,44,45]. Therefore, smoking habit may serve as a proxy indicator for lower adherence to preventive measures, as evidenced by the fact that current smokers were less inclined to utilise specific preventive services in comparison to non-smokers [51,52,53]. However, further research is needed to better understand the complex relationship between smoking habits, vaccine attitudes, and health outcomes.

Another result of this study was the association between being female and attitudes regarding vaccinations, which is consistent with previous research [21,33,54]. An explanation for this observed gender difference could be that females are generally more health-conscious [55]. This could lead to actively seeking health information and adhering to preventive health measures, including vaccination.

Another result of this study was the association between being single and attitudes. These findings are in contrast with previous studies, which showed that being single was linked to higher rates of vaccine hesitancy [56,57]. Therefore, it is important to note that single individuals may be particularly vulnerable to vaccine hesitancy due to their reduced exposure to social pressures [57]. To ensure that single individuals make informed decisions about vaccination, it is crucial to provide them with adequate support and information [57]. Such support can be instrumental in promoting better vaccination coverage, which is essential for safeguarding public health and preventing the spread of infectious diseases. It is, therefore, imperative that we reach out to single individuals and make them aware of the benefits of vaccination, while addressing their concerns and questions with accurate and transparent information.

Furthermore, this study revealed a noteworthy association between knowledge and attitude, suggesting that knowledge regarding vaccinations may contribute to the development of positive attitudes towards vaccinations. This finding is consistent with previous research [24,38,48,58], which emphasised the importance of accurate knowledge in fostering positive attitudes toward vaccinations. The significance of this relationship is particularly relevant to public health promotion and the prevention of VPDs. Thus, having access to reliable and precise information about vaccinations is critical in making decisions and cultivating positive attitudes toward vaccinations.

The results presented in Figure 3 suggest that women demonstrated slightly better behaviours. In contrast, a survey conducted in Italy during October 2009 found an interesting association between being male and exhibiting better behaviours towards preventive measures [59]. One possible explanation for the observed gender difference is that women tend to be more health-conscious and demonstrate a higher level of involvement in health-related practices [55].

One of the results of this study was the association between having children and better behaviours. This finding is consistent with previous research, which suggests that parents are more likely to adhere to vaccination recommendations for themselves and their children, feeling a greater responsibility to protect their children from VPDs and, therefore, possibly being more motivated to vaccinate themselves and their children [60,61].

Furthermore, this study revealed an association between non-smoking habits and behaviours regarding vaccinations. This finding is consistent with previous research that identified smoking as a barrier to vaccine uptake [42,43,44,45]. Overall, the association between non-smoking habits and positive behaviours towards vaccinations suggests that promoting healthy lifestyle behaviours such as not smoking may be an effective approach to increase vaccine uptake and reduce vaccine hesitancy.

This study’s findings emphasise the influential role of attitudes in shaping healthy behaviours, particularly in terms of vaccine acceptance. Individuals with positive attitudes towards vaccinations are more likely to engage in healthy behaviours, including accepting vaccinations. Furthermore, the presence of attitudinal barriers has been consistently observed in various studies conducted across different countries, such as Bangladesh [62] and Egypt [63]. These barriers encompass a range of factors, including lack of information about the vaccination and its adverse effects [63], limited access to vaccination coverage [64], affordability issues [65], and individual concerns regarding side effects and vaccine efficacy [66]. These findings further underscore the importance of attitudes in shaping healthy behaviours, as also emphasised in previous studies [67].

Another important finding from this study highlights the influence of knowledge on healthy behaviours. Individuals who lack knowledge about vaccinations are more susceptible to believing in misconceptions or false statements, such as the unfounded belief that the vaccine can cause the flu [68]. Similarly, misconceptions related to the safety of the vaccine for the fetus, such as an increased risk of miscarriage or birth defects, have been identified as a decisive barrier in pregnant women [69]. Interestingly, a study revealed that vaccine uptake increased among individuals who mistakenly believed that the vaccine protects against the common cold [43]. This indicates that certain misconceptions, even if inaccurate, can still have an impact on vaccine acceptance.

These results underscore the importance of addressing knowledge gaps as a crucial step in enhancing vaccine acceptance and promoting the adoption of healthy behaviours.

## 5. Limitation

This study had several limitations that need to be considered when interpreting its findings. One major limitation was the use of self-reported behaviours collected through questionnaires, which may have led to social desirability bias. However, the study attempted to mitigate this by ensuring anonymity and confidentiality for the participants.

Another limitation is the potential for selection bias, as certain individuals or groups may have been excluded from the study. Confounding bias is also a possibility if there were other variables that influenced the results but were not accounted for in the analysis.

Moreover, the study’s small sample size relative to the population of Naples limits the generalisability of the results. It is important to acknowledge that the findings may not be representative of the entire population.

Another limitation of this study lies in its primary emphasis on demographic classifications rather than conducting an in-depth analysis of the underlying causes of varying vaccine uptake rates. Furthermore, our research does not extensively investigate the mechanisms through which the studied population acquires knowledge about vaccination benefits and risks.

Additionally, the utilisation of snowball sampling may introduce selection biases. However, this method was used for the initial sampling; in a second phase, we extended the sampling through multiple channels to increase the representativeness of the sample.

Finally, while the use of a KAP-based questionnaire was a useful tool for capturing knowledge, attitudes, and practices related to vaccinations, it may not have captured all the factors that influence vaccine-related beliefs and behaviours. The cross-sectional design of the study restricts our ability to establish causality or capture temporal changes in KAP; due to these constraints, the questionnaire used in this study might have had a more comprehensive validation process that, over time, potentially may affect the reliability and validity of the instrument. Future studies could consider incorporating additional measures to address these limitations and provide a more comprehensive understanding of vaccine-related factors.

Overall, it is important to interpret the findings of this study within the context of these limitations and consider them when applying the results to other settings or populations.

## 6. Policies

The findings of this study underscore the substantial influence of knowledge and attitudes in fostering vaccine uptake. A higher level of knowledge and positive attitudes towards vaccinations are strongly associated with increased engagement in healthy behaviours, particularly regarding vaccine acceptance. Conversely, a lack of understanding about vaccinations increases susceptibility to misconceptions and false beliefs, while attitudinal barriers can hinder vaccine uptake.

Individuals who possess a deeper comprehension of vaccinations and their benefits are more likely to hold positive attitudes towards vaccination and adhere to vaccination plans. To effectively target population groups that are more prone to vaccine refusal, such as men, smokers, and individuals with lower educational levels, public health authorities should possess a comprehensive understanding of the social, demographic, and psychological determinants of vaccine hesitancy.

Language selection in health messages and the use of appropriate communication strategies and media also play a vital role in vaccine acceptance [26,27]. All health authorities involved in health communication must align their efforts to produce clear and coherent messages. Tailored strategies and evidence-based educational interventions should be developed to promote informed decision-making and increase vaccine uptake rates [70].

Current and future vaccination campaigns should highlight the central role of individuals, focus on improving health literacy, and encourage engagement with the healthcare system. These actions should effectively communicate the medical, scientific, and social value of vaccination while fostering trust, awareness, and responsible action [71].

By prioritising efforts to improve vaccine adherence and address vaccine hesitancy, these policies and interventions can help mitigate the risk of VPDs and their associated complications, thereby contributing to overall improvements in public health.

## 7. Conclusions

This cross-sectional survey examines the impact of demographic factors, educational attainment, knowledge, attitudes, and behaviours on vaccine acceptance and adherence. A 31-question questionnaire was administered to 1163 adult participants, and the data were analysed using descriptive statistics and MLRA. The findings underscore the significance of these factors in understanding and promoting vaccination behaviours.

Notably, non-smoking habits were strongly associated with vaccine knowledge, attitudes, and behaviours. Additionally, being single and having a university degree were identified as indicators of higher knowledge and more positive attitudes. This study emphasises the pivotal role of vaccine knowledge in shaping positive attitudes and encouraging favourable behaviours. Furthermore, being female, having children, and exhibiting positive attitudes were linked to potential improvements in vaccine-related behaviours.

These results are of great importance as they focus on increasing knowledge, fostering positive attitudes, and promoting desirable behaviours regarding vaccinations. They serve as valuable indicators of the broader influences on vaccine-related outcomes, encouraging future endeavours to enhance vaccine acceptance and adherence.

## Figures and Tables

**Figure 1 vaccines-11-01558-f001:**
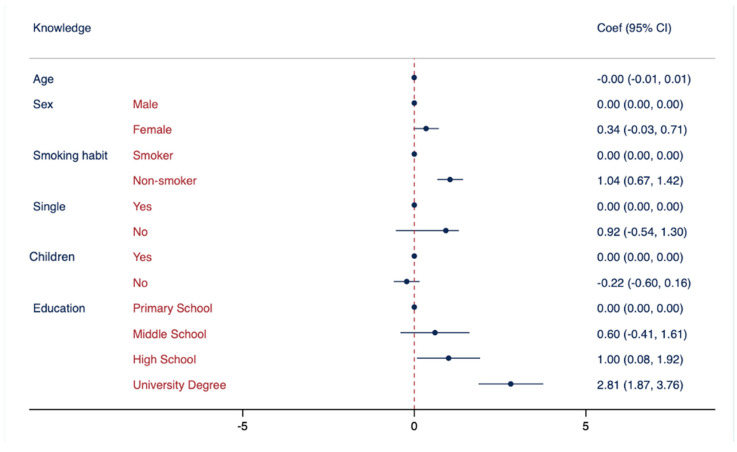
Association between knowledge regarding vaccinations and demographic characteristics. Multivariate logistic regressions were employed, including knowledge regarding vaccinations as the outcome variable and controlled for the following variables: age, sex, smoking habits, marital status, having children, and education attainment. Results are presented as Coef. and CI.

**Figure 2 vaccines-11-01558-f002:**
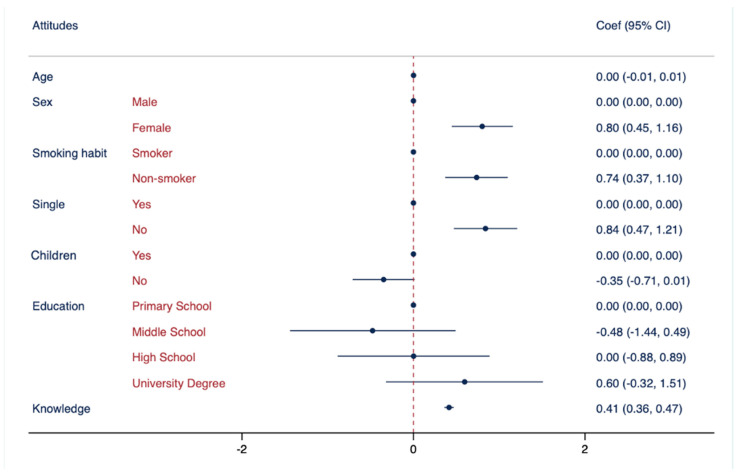
Association between attitude regarding vaccinations and demographic characteristics. Multivariate logistic regressions were employed, including attitude regarding vaccinations as the outcome variable and controlled for the following variables: age, sex, smoking habits, marital status, having children, education attainment, and knowledge. Results are presented as Coef. and CI.

**Figure 3 vaccines-11-01558-f003:**
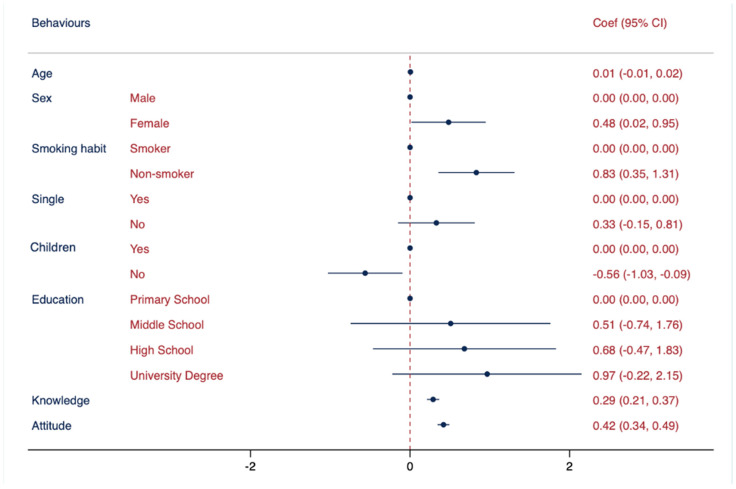
Association between behaviours regarding vaccinations and demographic characteristics. Multivariate logistic regressions were employed, including behaviours regarding vaccinations as the outcome variable and controlled for the following variables: age, sex, smoking habits, marital status, having children, education attainment, knowledge, and attitude. Results are presented as Coef. and CI.

**Table 1 vaccines-11-01558-t001:** Study population characteristics.

Study Population	N	Percentage
**Sex**	1163	
Male	535	46.00
Female	628	54.00
**Age**		
<30	281	24.16
31–50	414	35.60
>51	468	40.24
**Education**		
Primary school	51	4.39
Middle school	166	14.27
High school	569	48.93
University Degree	377	32.42
**Children**		
Yes	602	51.68
No	562	48.32
**Smoking habits**		
Yes	512	44.02
No	651	55.98
**Marital Status**		
Single	482	41.44
In a relationship	681	58.56

**Table 2 vaccines-11-01558-t002:** Knowledge of respondents regarding vaccinations.

N.	Statement (*Variables*)	Agree (%)	Uncertain (%)	Disagree (%)
K1	Antibodies are molecules produced by cells in the human body.	50.04 *	23.65	26.31
K2	Antibodies bind and destroy microorganisms.	38.52 *	27.69	33.79
K3	Vaccinations stimulate antibody production.	44.97 *	30.70	24.33
K4	There are currently ten compulsory vaccinations in total in Italy.	48.24 *	28.72	23.04
K5	There are currently four compulsory vaccinations in total in Italy.	31.21	34.74	34.05 *
K6	Tetanus vaccine booster is recommended in adulthood, every ten years.	39.55 *	31.81	28.63
K7	The hepatitis B vaccine is mandatory.	46.52 *	28.80	24.68
K8	The HPV vaccine is only recommended in girls.	24.51	28.63	46.86 *
K9	3 in 100,000 doses administered have serious adverse vaccination reactions.	32.67 *	43.25	24.08
K10	30 in 100,000 doses administered have serious adverse vaccination reactions.	26.14	39.72	34.14 *
K11	Failure to administer vaccinations prohibits enrolment in school according to D.Lgs. 73/17 (Italian regulation regarding mandatory vaccination issued by the Ministry of Health).	36.46 *	30.87	32.67

***** Correct answer.

**Table 3 vaccines-11-01558-t003:** Attitude of respondents toward vaccinations.

N.	Statement (*Variables*)	Agree (%)	Uncertain (%)	Disagree (%)
A1	You must always have everything under control.	33.88	25.54	40.58
A2	Vaccinations are just a business.	24.25	26.23	49.53
A3	It is preferable to use natural remedies instead of drugs.	39.38	26.74	33.88
A4	Health records must be kept.	53.57	24.25	22.18
A5	Better safe than sorry.	51.50	25.88	22.61
A6	Vaccinations are more dangerous than infections.	27.34	28.46	44.20
A7	It is essential to sanitise hands.	41.44	32.67	25.88
A8	It is necessary for the treating doctor to practice alternative medicine.	30.95	28.72	40.33
A9	The only reason I vaccinate my child is so they can go to kindergarten or school.	28.20	26.05	45.74
A10	The syringes are scary.	28.55	23.73	47.72

**Table 4 vaccines-11-01558-t004:** Behaviours of respondents concerning vaccinations.

N.	Questions	Yes/Always (%)	Often (%)	Sometimes (%)	Never (%)
B1	You have had the recommended vaccinations?	30.09	26.83	20.38	22.70
B2	Do you keep health records?	41.53	18.57	20.46	19.43
B3	Do you participate/have you ever participated in vaccination campaigns?	33.96	25.11	19.69	21.24
B4	Do you get vaccine boosters?	39.55	20.81	21.32	18.31
B5	Do you inquire about vaccinations when you travel?	33.10	25.62	21.84	19.43
B6	Do you get a flu shot every year?	34.31	20.29	21.15	24.25
B7	Have you been vaccinated and/or vaccinated your son/daughter against Papilloma Virus?	29.32	19.35	22.96	28.37
B8	Have you ever been tested for hepatitis b antibodies?	26.23	28.20	19.17	26.40
B9	Do you inform yourself about the risks of vaccinations?	37.92	21.07	20.03	20.98
B10	After a skin wound, do you get a tetanus shot?	25.88	16.51	25.02	32.59

**Table 5 vaccines-11-01558-t005:** Results of the linear multiple regression analysis (MLRA).

	Coefficients Not Standardised		Coefficients Standardised			
	b	Standard Error	t	95% Conf. Interval		*p*-Value
**Model I–Dependent variable: Knowledge**						
*Prob > F = 0.000*	*R-squared = 0.1414*			*Root-MSE = 3.2094*		
Age	−0.004	0.006	−0.67	−0.015	0.007	0.504
Sex	0.342	0.190	1.81	−0.029	0.715	0.071
Marital status	0.920	0.194	4.74	0.539	1.30	0.000
Children	−0.221	0.192	−1.15	−0.599	0.156	0.250
Smoking habits	1.04	0.192	5.44	0.668	1.42	0.000
Education *						
Middle School	0.603	0.515	1.17	−0.407	1.61	0.242
High School	1.00	0.470	2.13	0.077	1.92	0.034
University Degree	2.81	0.481	5.85	1.87	3.76	0.000
**Model II–Dependent variable: Attitudes**						
*Prob > F = 0.000*	*R-squared = 0.2771*			*Root-MSE = 3.0755*		
Age	0.001	0.005	0.12	−0.010	0.011	0.906
Sex	0.802	0.182	4.41	0.446	1.16	0.000
Marital status	0.840	0.188	4.47	0.471	1.21	0.000
Children	−0.348	0.184	−1.89	−0.710	0.013	0.059
Smoking habits	0.737	0.186	3.95	0.371	1.10	0.000
Education *						
Middle School	−0.477	0.493	−0.97	−1.44	0.492	0.334
High School	0.002	0.452	0.00	−0.884	0.888	0.997
University Degree	0.597	0.468	1.28	−0.321	1.51	0.202
Knowledge	0.415	0.028	14.7	0.360	0.470	0.000
**Model III–Dependent variable: Behaviour**						
*Prob > F = 0.000*	*R-squared = 0.2840*			*Root-MSE = 3.979*		
Age	0.005	0.007	0.72	−0.009	0.019	0.474
Sex	0.483	0.237	2.03	0.017	0.949	0.042
Marital status	0.330	0.245	1.35	−0.151	0.811	0.178
Children	−0.563	0.239	−2.36	−1.03	−0.094	0.019
Smoking habits	0.831	0.243	3.42	0.355	1.31	0.001
Education *						
Middle School	0.509	0.639	0.80	−0.745	1.76	0.426
High School	0.681	0.584	1.17	−0.466	1.83	0.244
University Degree	0.967	0.606	1.60	−0.221	2.15	0.111
Knowledge	0.289	0.040	7.26	0.211	0.367	0.000
Attitude	0.419	0.038	10.9	0.344	0.493	0.000

***** Primary School as Reference. References: “male” as the reference for sex, “smoker” as the reference for smoking habits, “being single” as the reference for marital status, “have children” as the reference for having children, and “primary school” as the reference for education attainment.”

## Data Availability

Not applicable.

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
