# Peer review of "Understanding Factors Contributing to Vaccine Hesitancy in a Large Metropolitan Area"

_vaccines, 2023, doi:10.3390/vaccines11101558_

Round 1

Reviewer 1 Report

This is an interesting and generally well-written manuscript. There are a few changes I would recommend.

For the knowledge items, it would be helpful for the Table to give the right answers and indicate the % of respondents who got the right answers (and not just the % who agreed).

Currently some new analysis is introduced in the discussion section. This needs to be moved to the results section and then discussed (as needed) in the discussion section. The discussion shouldn't contain new data or analysis.

The main weakness of the study is the snowballing method of recruitment. While this method can accelerate the rate of recruitment, two major problems are (i) the loss of sampling control by the researchers and (ii) the probability that existing respondents recommend new potential respondents who are similar to themselves. This second point is a particular concern in a study of attitudes and behaviours that are socially patterned, as the case with vaccinations. It seems odd that the researchers chose this method: snowballing is legitimate and can be useful, but it is most often used when respondents are needed with a characteristic that is either rare or unknown to researchers or socially stigmatised. That isn't the case with this study - the researchers wanted a sample of adults from the general Italian population. The main weakness of the snowballing method is that the application of the results to the wider population isn't possible. These recruitment and sampling factors are weaknesses and, although they don't entirely invalidate the study results, they should be acknowledged as such within the paper.

Mostly very good. In a few places minor changes could be made. 

Author Response

This is an interesting and generally well-written manuscript. There are a few changes I would recommend.

  • For the knowledge items, it would be helpful for the Table to give the right answers and indicate the % of respondents who got the right answers (and not just the % who agreed).

Thank you for your feedback. We acknowledge that the reviewer's suggestion provides more accurate information about the content of Table 1. To this end, we have added asterisks to indicate correct answers, which also indicate the percentage of participants who answered correctly, and the statement "* correct answer" as a caption in Table I.  

  • Currently some new analysis is introduced in the discussion section. This needs to be moved to the results section and then discussed (as needed) in the discussion section. The discussion shouldn't contain new data or analysis.

Thank you for highlighting the discrepancy between the 'Results' and 'Discussion' paragraphs. Your input has been invaluable in refining the quality and comprehensibility of our manuscript. We acknowledge that the “Results” paragraph should provide more specific details about the associations observed in each model, and we have made the necessary adjustments in lines 186-191. Also, we would like to clarify that in our analysis, we utilized in each model specific reference for the categories: specifically, we employed 'male' as the reference category for sex, 'smoker' as the reference category for smoking habits, 'being single' as the reference category for marital status, 'have children' as the reference category for having children, and 'primary school' as the reference category for education attainment. To ensure transparency and facilitate interpretation, we will add this information as a caption in Table V, providing a clear point of comparison for the associations observed in each model. Furthermore, we will include the following statement in lines 185-188 to explicitly mention the reference categories used: “In each model, specific references were assigned to the categories: “male” as reference for sex, “smoker” as reference for smoking habits, “being single” as reference for marital status, “have children” as reference for having children, and “primary school” as reference for education attainment.”Furthermore, during our re-evaluation of the article, we identified a transcriptional error in the 'Results' section, specifically in Model II, where there was an incorrect association mentioned between attitudes and education attainment. We apologize for this oversight and have promptly corrected the error.

These changes aim to enhance the transparency and comprehensibility of our manuscript. Thank you again for your valuable feedback.

  • The main weakness of the study is the snowballing method of recruitment. While this method can accelerate the rate of recruitment, two major problems are (i) the loss of sampling control by the researchers and (ii) the probability that existing respondents recommend new potential respondents who are similar to themselves. This second point is a particular concern in a study of attitudes and behaviours that are socially patterned, as the case with vaccinations. It seems odd that the researchers chose this method: snowballing is legitimate and can be useful, but it is most often used when respondents are needed with a characteristic that is either rare or unknown to researchers or socially stigmatised. That isn't the case with this study - the researchers wanted a sample of adults from the general Italian population. The main weakness of the snowballing method is that the application of the results to the wider population isn't possible. These recruitment and sampling factors are weaknesses and, although they don't entirely invalidate the study results, they should be acknowledged as such within the paper.

Thank you for your thoughtful comment. Addressing your concerns, we would like to provide a more detailed explanation of the snowball sampling method used in our research.

In our study, the snowball sampling method was employed to identify and recruit the initial participants. We initially asked to our trusted trial group, as indicated in the methods section, to indicate 2-3 subject to form a diverse group of individuals from the general population in the metropolitan area that might be interested in participating at the study. Then we approached potential participants through community organizations, local health clinics, and social media platforms to ensure a varied and representative sample. However, to better clarify this methodology and guarantee the reproducibility and transparency of the paper, we decided to add this paragraph at the “Method” section, lines 81-87, to better clarify how the recruitment was made: “The initial participants were identified and recruited using a snowball sampling method, starting with a trusted trial group recommending 2-3 subjects who represented a diverse selection from the general population within the metropolitan area and had expressed interest in participating in the study. Subsequently, we reached out to potential participants through multiple channels, including community organizations, local health clinics, and social media platforms, to ensure a varied and representative sample”. Also, we concur that the sampling method has some inherent limitations. To address these points and provide greater transparency, we will include a statement in the limitations section,lines 363-366, as follows: “Additionally, the utilization of snowball sampling, which may introduce selection biases. However, this method, was used for the initial sampling, in a second phase we extended the sampling through multiple channels to increase the representativeness of the sample.”.

The utilization of the snowball sampling method in this study aligns with prior research endeavors, including studies conducted by both our university and other research groups, all of which focused on the general population. This affirms its suitability for our study's objectives and methodology:

- P. Montuori et al., “Determinants Analysis Regarding Household Chemical Indoor Pollution,” Toxics, vol. 11, no. 3, p. 264, Mar. 2023, doi: 10.3390/toxics11030264.

- F. Pennino et al., “Understanding Hearing Health: A Cross-Sectional Study of Determinants in a Metropolitan Area,” Healthcare, vol. 11, no. 16, p. 2253, Aug. 2023, doi: 10.3390/healthcare11162253.

- P. Montuori et al., “Assessment on Practicing Correct Body Posture and Determinant Analyses in a Large Population of a Metropolitan Area,” Behavioral Sciences, vol. 13, no. 2, p. 144, Feb. 2023, doi: 10.3390/bs13020144.

- Reuben, R.C., Danladi, M.M.A., Saleh, D.A. et al. Knowledge, Attitudes and Practices Towards COVID-19: An Epidemiological Survey in North-Central Nigeria. J Community Health 46, 457–470 (2021). https://doi.org/10.1007/s10900-020-00881-1

- Winarto H, Habiburrahman M, Dorothea M, Wijaya A, Nuryanto KH, et al. (2022) Knowledge, attitudes, and practices among Indonesian urban communities regarding HPV infection, cervical cancer, and HPV vaccination. PLOS ONE 17(5): e0266139. https://doi.org/10.1371/journal.pone.0266139

- Akel, M.; Sakr, F.; Haddad, C.; Hajj, A.; Sacre, H.; Zeenny, R.M.; Safwan, J.; Salameh, P. Knowledge, Attitude, and Practices of the General Population toward the Old-New Outbreak of Cholera in a Developing Country. Trop. Med. Infect. Dis. 2023, 8, 236. https://doi.org/10.3390/tropicalmed8040236

We appreciate your feedback, which has contributed to enhancing the transparency and comprehensibility of our research methodology.

Reviewer 2 Report

Please review my evaluation and comments contained in the attached file 

Review, evaluation, and comments:

Vaccines

 Understanding Factors Contributing to Vaccine Hesitancy in a Large Metropolitan Area

Paolo Montuori 1, Immanuela Gentile 1, Claudio Fiorilla 1, Michele Sorrentino 1, Benedetto Schiavone 1, 4 Valerio Fattore 1, Fabio Coscetta 1, Alessandra Riccardi 1, Antonio Villani 1, Ugo Trama 2, Francesca Pennino 1,*, 5 Maria Triassi 1 and Antonio Nardone 1

General comments: Comments by the reviewer are made with the highest respect for the research group.

The manuscript is well written by an experienced Italian group with many contributors. They all are members of the Public Health organizations situated in Naples Italy.

The study uses the standard and established KAP questionnaire platform sampling a small segment of the population listed at close to 1 million residents.

The sampling occurred over 3 months -Dec 2022 through Feb 2023- A time when all were at the tail end of the COVID pandemic.

The study is cross sectional and proposes to expand the objective to determine KAP for general vaccinations not specifically focused of COVID-19 vaccine uptake.

The overall study methodology and statistical analysis appears to be established and consistent with public health research practices.

Specific comments:

1.     Population sampling – It’s unclear from the description on how the subjects were identified within the Naples area as potential subjects. Was the survey/ project advertised on media social media emails etc. As stated in their limitations this could have introduced serious limitations related to a biased population willing to participate. More details on this recruitment strategy should be forth coming. The snowball sampling technique noted should have a reference and described in more detail. 

2.     Sample size- this was stated but I am unsure what specific primary outcome effect size was used for these calculations. Needs more detail.

3.     Human studies protection- It doesn’t appear that the authors' obtained an IRB review evaluation ruling of the category of project such as exempt expedited non research etc. The investigators have made their own decision that the project does not require IRB review. It is standard at least in the US for all human interaction projects that are not quality improvement for local use -including de-identified survey projects to have been reviewed for human subject research protection issues.  A statement specific as to why this was not done would be helpful.

4.     Questionnaire development- The KAP approach is quite standard. However, the reviewer is concerned on the steps taken in developing and validating the 3 KAP domains. There are established steps in developing a questionnaire that has internal and external – face, content, and construct validity. It appears only one step was taken that is subjecting the content questions for comprehension after the research team developed the set of questions. Granted so many surveys in the literature are unvalidated but the authors should provide additional information-rationale as to why a more rigorous course for questionnaire validation was not pursued and how that would not impact the outcomes.

5.     Responses- do the authors feel the chosen response choices are adequately understood by the subjects? Does “often” mean > 50% or > 75% for example or does it really make any difference in the outcomes.

6.     Knowledge questions- I had some concerns related to the questions as it relates directly to a subject understanding of vaccination knowledge. The statement of “antibodies binding and destroying micro-organisms” it not strictly correct. I understand the need to provide a generic question, but this question may not be as discriminating as required. Perhaps a subject that is a bit more sophisticated would answer disagree whereas one with less knowledge would agree but both would be correct. Questions K4-K9 seem more in line with vaccination policy and requirements and not necessarily related to how a subject understands vaccination and biological and safety effects.

K9 and K10 may have too specific as to the numbers and probably led to subject “guessing”. A more qualitative type of assessment question might have more validity.

7.     Behavior questions- B3- does the subject understand what a campaign is and how is it relevant? B4- “do you do vaccine recalls”- not sure what this means? B9- Is “fester” understood by all Italians as to the intent or is this a translation issue? I know the project was related to vaccinations in general but why wasn’t there a specific question on receiving the COVID vaccine and boosters?

8.     Results- In general the results are interesting, and some are novel, but many reinforce previous findings from earlier studies. With the sampling limitations the results do suggest potential important leads in future vaccination campaigns within the Naples area but uncertain they can be translated to other areas of Italy or beyond. These limitations were described well in the manuscript.

Unfortunately, the study does not tease out roots causes for the low or high vaccine uptake specifically but rather classifies based on demographics. The other challenge is understanding how Italians gain knowledge or not related to the benefits and risks of vaccinations as that stands as a major point for intervention. The other unmet need is understanding how public health entities are providing knowledge enhancing interventions in general for vaccinations both to the public and health care providers. 

Author Response

Specific comments:

  • Population sampling – It’s unclear from the description on how the subjects were identified within the Naples area as potential subjects. Was the survey/ project advertised on media social media emails etc. As stated in their limitations this could have introduced serious limitations related to a biased population willing to participate. More details on this recruitment strategy should be forth coming. The snowball sampling technique noted should have a reference and described in more detail.  

Thank you for your thoughtful comment. Addressing your concerns, we would like to provide a more detailed explanation of the snowball sampling method used in our research.

In our study, the snowball sampling method was employed to identify and recruit the initial participants. We initially asked to our trusted trial group, as indicated in the methods section, to indicate 2-3 subject to form a diverse group of individuals from the general population in the metropolitan area that might be interested in participating at the study. Then we approached potential participants through community organizations, local health clinics, and social media platforms to ensure a varied and representative sample. However, to better clarify this methodology and guarantee the reproducibility and transparency of the paper, we decided to add this paragraph at the “Method” section, lines 81-87, to better clarify how the recruitment was made: “The initial participants were identified and recruited using a snowball sampling method, starting with a trusted trial group recommending 2-3 subjects who represented a diverse selection from the general population within the metropolitan area and had expressed interest in participating in the study. Subsequently, we reached out to potential participants through multiple channels, including community organizations, local health clinics, and social media platforms, to ensure a varied and representative sample”. Also, we concur that the sampling method has some inherent limitations. To address these points and provide greater transparency, we will include a statement in the limitations section,lines 363-366, as follows: “Additionally, the utilization of snowball sampling, which may introduce selection biases. However, this method, was used for the initial sampling, in a second phase we extended the sampling through multiple channels to increase the representativeness of the sample.”.

The utilization of the snowball sampling method in this study aligns with prior research endeavors, including studies conducted by both our university and other research groups, all of which focused on the general population. This affirms its suitability for our study's objectives and methodology:

- P. Montuori et al., “Determinants Analysis Regarding Household Chemical Indoor Pollution,” Toxics, vol. 11, no. 3, p. 264, Mar. 2023, doi: 10.3390/toxics11030264.

- F. Pennino et al., “Understanding Hearing Health: A Cross-Sectional Study of Determinants in a Metropolitan Area,” Healthcare, vol. 11, no. 16, p. 2253, Aug. 2023, doi: 10.3390/healthcare11162253.

- P. Montuori et al., “Assessment on Practicing Correct Body Posture and Determinant Analyses in a Large Population of a Metropolitan Area,” Behavioral Sciences, vol. 13, no. 2, p. 144, Feb. 2023, doi: 10.3390/bs13020144.

- Reuben, R.C., Danladi, M.M.A., Saleh, D.A. et al. Knowledge, Attitudes and Practices Towards COVID-19: An Epidemiological Survey in North-Central Nigeria. J Community Health 46, 457–470 (2021). https://doi.org/10.1007/s10900-020-00881-1

- Winarto H, Habiburrahman M, Dorothea M, Wijaya A, Nuryanto KH, et al. (2022) Knowledge, attitudes, and practices among Indonesian urban communities regarding HPV infection, cervical cancer, and HPV vaccination. PLOS ONE 17(5): e0266139. https://doi.org/10.1371/journal.pone.0266139

- Akel, M.; Sakr, F.; Haddad, C.; Hajj, A.; Sacre, H.; Zeenny, R.M.; Safwan, J.; Salameh, P. Knowledge, Attitude, and Practices of the General Population toward the Old-New Outbreak of Cholera in a Developing Country. Trop. Med. Infect. Dis. 2023, 8, 236. https://doi.org/10.3390/tropicalmed8040236

We appreciate your feedback, which has contributed to enhancing the transparency and comprehensibility of our research methodology.

  • Sample size-this was stated but I am unsure what specific primary outcome effect size was used for these calculations. Needs more detail. 

The sampling method we utilized was a simple random sample, calculated using Slovin’s formula to ensure the representativeness of our sample. Following this calculation, we extended invitations to 1670 individuals, ultimately resulting in the participation of 1163 subjects. This approach was chosen to maintain a margin of error within 3% and a 95% confidence interval, thereby ensuring the representativeness of our study population. Our primary objective was to assess vaccine hesitancy within the general population. It's crucial to note that vaccine hesitancy is a complex, population-wide phenomenon and therefore we decided to not use a specific primary outcome to calculate the size sample.

  • Human studies protection-It doesn’t appear that the authors' obtained an IRB review evaluation ruling of the category of project such as exempt expedited non research etc. The investigators have made their own decision that the project does not require IRB review. It is standard at least in the US for all human interaction projects that are not quality improvement for local use -including de-identified survey projects to have been reviewed for human subject research protection issues.  A statement specific as to why this was not done would be helpful. 

In terms of ethical considerations, participation in the questionnaire was entirely voluntary, and we guaranteed anonymity to all participants. In the Materials and Methods Section, under paragraph “2.2. Procedures,” we clarified the following: "Participants were provided with comprehensive information about the nature, scope, and methodology of the research. Prior consent was obtained from participants, and they were informed that they had the right to withdraw their consent and discontinue participation at any time without providing a reason. The present study was conducted in accordance with the guidelines and principles outlined in the Declaration of Helsinki.”

Furthermore, in accordance with local and national legislation, this type of study does not require direct approval from the ethics committee, as stipulated in art. 12, paragraph 10, letter c) of D. Lgs. n. 158 of 13 September 2012, converted with amendments in Law No. 189 of 8 November 2012, and DM 8 February 2013. To provide further clarity on this matter, we have included the statement, line 110-111, “ethical clearance was obtained according to local legislation.”

These revisions and clarifications align with the Guidelines for Authors, which state that, “For non-interventional studies (e.g., surveys, questionnaires, social media research), all participants must be fully informed if anonymity is assured, why the research is being conducted, how their data will be used, and if there are any risks associated. As with all research involving humans, ethical approval from an appropriate ethics committee must be obtained prior to conducting the study. If ethical approval is not required, authors must either provide an exemption from the ethics committee or are encouraged to cite the local or national legislation that indicates ethics approval is not required for this type of study. Where a study has been granted exemption, the name of the ethics committee which provided this should be stated in Section ‘Institutional Review Board Statement’ with a full explanation regarding why ethical approval was not required.”

  • Questionnaire development-The KAP approach is quite standard. However, the reviewer is concerned on the steps taken in developing and validating the 3 KAP domains. There are established steps in developing a questionnaire that has internal and external – face, content, and construct validity. It appears only one step was taken that is subjecting the content questions for comprehension after the research team developed the set of questions. Granted so many surveys in the literature are unvalidated but the authors should provide additional information-rationale as to why a more rigorous course for questionnaire validation was not pursued and how that would not impact the outcomes.

We appreciate the reviewer’s attention to the questionnaire development process and understand the importance of robust validation steps for questionnaires. In our study, we acknowledge that a more extensive validation process could have been pursued, but several factors influenced our decision to employ a less rigorous approach.

Our research’s scope and available resources significantly influenced our approach to questionnaire validation. Given constraints related to time and budget, we chose a more practical approach to ensure the study’s feasibility. Conducting a comprehensive validation process would have required additional time and resources that were not available for this project.

Our questionnaire development involved a team of experts in the field, including subject matter specialists and experienced researchers. Their contributions were pivotal in formulating relevant, comprehensive questions that effectively covered the three domains of knowledge, attitude, and behaviors. In addition to individual question comprehension checks, we conducted pilot testing of the entire questionnaire with a small group of participants. This step helped identify and address ambiguities, inconsistencies, or issues related to question phrasing, further enhancing the questionnaire’s quality.

Although our specific questionnaire had not previously undergone validation in the literature, it’s essential to note that many individual questions were adapted from validated instruments used in similar studies, which are listed below. This adaptation significantly ensured the face validity of our questionnaire.

  • Fernandes A, Wang D, Domachowske JB, Suryadevara M. Vaccine knowledge, attitudes, and recommendation practices among health care providers in New York State. Hum Vaccin Immunother. 2023 Dec 31;19(1):2173914. doi: 10.1080/21645515.2023.2173914. Epub 2023 Feb 7. PMID: 36749617; PMCID: PMC10026857.
  • Riccò M, Vezzosi L, Gualerzi G, Signorelli C. Knowledge, attitudes and practices (KAP) towards vaccinations in the school settings: an explorative survey. J Prev Med Hyg. 2017 Dec 30;58(4):E266-E278. doi: 10.15167/2421-4248/jpmh2017.58.4.673. PMID: 29707657; PMCID: PMC5912794.
  • Bénédicte Melot, Paola Bordin, Caterina Bertoni, Valentina Tralli, Mariagrazia Zuccali, Andrea Grignolio, Silvia Majori & Antonio Ferro(2021) Knowledge, attitudes and practices about vaccination in Trentino, Italy in 2019, Human Vaccines & Immunotherapeutics, 17:1, 259-268, DOI: 1080/21645515.2020.1763085
  • Yutong Jiang, Xi Zhang, Qing Lv, Jun Qi, Xinghua Guo, Qiujing Wei, Zetao Liao, Zhiming Lin & Jieruo Gu(2019)Knowledge, attitude, and practice regarding infection and vaccination in patients with rheumatic diseases in China,Human Vaccines & Immunotherapeutics, 15:5, 1100-1105, DOI: 1080/21645515.2019.1568160
  • Riccò, M.; Peruzzi, S. Tetanus Vaccination Status and Vaccine Hesitancy in Amateur Basketball Players (Italy, 2020).Vaccines 2022, 10, 131. https://doi.org/10.3390/vaccines10010131
  • Zakhour R, Tamim H, Faytrouni F, Khoury J, Makki M, Charafeddine L. Knowledge, attitude and practice of influenza vaccination among Lebanese parents: A cross-sectional survey from a developing country. PLoS One. 2021 Oct 14;16(10):e0258258. doi: 10.1371/journal.pone.0258258. PMID: 34648535; PMCID: PMC8516244.
  • Alexandra Zingg, Michael Siegrist. Measuring people's knowledge about vaccination: Developing a one-dimensional scale. Vaccine, Volume 30, Issue 25,2012, Pages 3771-3777, ISSN 0264-410X, https://doi.org/10.1016/j.vaccine.2012.03.014.

Our primary goal was to provide timely insights into the Knowledge, Attitudes, and Practices (KAP) of a specific population, with potential implications for public health decision-making. We believed that the benefits of promptly available information outweighed the potential drawbacks of a more time-consuming validation process, which could have led to research delays.

We maintain that the steps we undertook, including expert input, pilot testing, and leveraging established questions, collectively contribute to the validity and reliability of our questionnaire. Our commitment to transparency remains unwavering, and we have taken measures to clearly describe the questionnaire development process in our manuscript.

In light of these considerations and to ensure transparency, we will incorporate a statement, lines 369-375, in the limitations section of our manuscript to address these points. We remain committed to providing valuable insights into the KAP of our target population, with a full awareness of the constraints under which our study was conducted:

"The cross-sectional design of the study that restricts our ability to establish causality or capture temporal changes in KAP, due to these constraints, the questionnaire used in this study might have had a more comprehensive validation process that over time potentially may affect the reliability and validity of the instrument.”

  • Responses-do the authors feel the chosen response choices are adequately understood by the subjects? Does “often” mean > 50% or > 75% for example or does it really make any difference in the outcomes.

We do acknowledge that the 4-response option may not frame precisely the spectrum of individual behaviours as any other possible Lickert scale approach, however, is a necessary step to quantify a variable. In this case the response options used in the questionnaire were explained to the subjects before its administration by the interviewers. Specifically, "never" corresponded to 0%, "sometimes" to responses between 25% and 50%, "often" represented responses between 50% and 75%, and "always" referred to responses between 75% and 100%. These explanations aimed to ensure that the subjects understood the response categories and could provide accurate and were provided by the interviewers as part of the introductory instructions, aimed at fostering a common understanding among participants regarding the response categories and facilitating accurate and consistent responses across the study.

  • Knowledge questions- I had some concerns related to the questions as it relates directly to a subject understanding of vaccination knowledge. The statement of “antibodies binding and destroying micro-organisms” it not strictly correct. I understand the need to provide a generic question, but this question may not be as discriminating as required. Perhaps a subject that is a bit more sophisticated would answer disagree whereas one with less knowledge would agree but both would be correct. Questions K4-K9 seem more in line with vaccination policy and requirements and not necessarily related to how a subject understands vaccination and biological and safety effects.

We appreciate your feedback regarding the questions related to vaccination knowledge. The statement, "antibodies binding and destroying micro-organisms," was intended to be a simplified and generic representation of the immune response triggered by vaccination. We acknowledge that it may not capture the full complexity of the process. However, our aim with this question was to gauge a basic understanding of how vaccinations work. Questions K4-K9, which relate to vaccination policy and requirements, were included to assess participants' awareness of practical aspects of vaccination, such as the obligatory nature of certain vaccinations and the recommended vaccination frequencies. These questions complemented the broader understanding of vaccination processes. We designed the questionnaire with a combination of knowledge-related questions, including both basic and practical aspects, to provide a comprehensive assessment of participants' knowledge and awareness related to travel-related infectious diseases and vaccinations.

  • K9 and K10 may have too specific as to the numbers and probably led to subject “guessing”. A more qualitative type of assessment question might have more validity. 

Thank you for your input regarding questions K9 and K10. We understand your concern about the specificity of numerical questions and the possibility of respondents guessing their answers. However, the percentage of answers provided in response to these questions suggests that respondents were providing considered responses rather than guessing. The inclusion of these specific numerical questions aimed to gather data on awareness regarding vaccination availability and awareness, which can be valuable in understanding the depth of respondents' knowledge on these topics.

  • Behavior questions- B3- does the subject understand what a campaign is and how is it relevant? B4- “do you do vaccine recalls”- not sure what this means? B9- Is “fester” understood by all Italians as to the intent or is this a translation issue? I know the project was related to vaccinations in general but why wasn’t there a specific question on receiving the COVID vaccine and boosters? 

Thank you for your input regarding questions B3:  while it's true that the term "campaign" may vary in familiarity among respondents, it's reasonable to assume a basic understanding, especially given the significant impact of the COVID-19 pandemic on the general population. The widespread use of the term "campaign" in relation to COVID-19 vaccination efforts likely increased its recognition. Given the focus on vaccine hesitancy in the research paper, understanding respondents' participation in vaccination campaigns is relevant because it provides insights into their engagement with vaccination-related initiatives. This information can help researchers gauge the extent to which individuals are actively involved in or aware of efforts to promote vaccination, which can, in turn, inform discussions on vaccine hesitancy and public health strategies.

Regarding B4 and B9, we appreciate your feedback. The questionnaire was originally designed and written in Italian for the article it was translated, and in this case in a misleading way. We rectified it as follows: B4 – “"Do you get vaccine boosters?"; B9 – “Do you inform yourself about the risks of vaccinations?"

Regarding specific question on COVID vaccine and boosters, we chose not to include specific questions about receiving the COVID vaccine and boosters in our survey due to the widespread vaccine hesitancy that had emerged across the country. Including such questions might have introduced complications with establishing a clear baseline for our study, as the context and sentiment regarding COVID vaccines were evolving rapidly. Our focus was primarily on broader vaccine hesitancy and knowledge of vaccines in general to ensure the study's stability and relevance over time.

  • Results- In general the results are interesting, and some are novel, but many reinforce previous findings from earlier studies. With the sampling limitations the results do suggest potential important leads in future vaccination campaigns within the Naples area but uncertain they can be translated to other areas of Italy or beyond. These limitations were described well in the manuscript. Unfortunately, the study does not tease out roots causes for the low or high vaccine uptake specifically but rather classifies based on demographics. The other challenge is understanding how Italians gain knowledge or not related to the benefits and risks of vaccinations as that stands as a major point for intervention. The other unmet need is understanding how public health entities are providing knowledge enhancing interventions in general for vaccinations both to the public and health care providers. 

We appreciate the reviewer's insightful comments regarding our study.  We concur that our study primarily focuses on demographic classifications rather than delving into the root causes of low or high vaccine uptake. To address these points and provide greater transparency, we will include a statement in the limitations section, lines 358-362, as follows: "Another limitation of this study lies in its primary emphasis on demographic classifications rather than conducting an in-depth analysis of the underlying causes of varying vaccine uptake rates. Furthermore, our research does not extensively investigate the mechanisms through which the studied population acquires knowledge about vaccination benefits and risks.". This statement will enhance the transparency of our study and guide future research endeavors in the field of vaccine-related behaviors and interventions.

Round 2

Reviewer 2 Report

The authors have offered both complete and rigorous responses to my comments and concerns that are excellent and address all issues 

I complement the group on their important efforts in public health